# SENSES-ASD: a social-emotional nurturing and skill enhancement system for autism spectrum disorder

Haya Abu-Nowar[1,*], Adeeb Sait[1,*], Tawfik Al-Hadhrami[1], Mohammed Al-Sarem[2] and Sultan Noman Qasem[3]

[1] Computer Science Department, School of Science and Technology, Nottingham Trent University, Nottingham, United Kingdom
[2] College of Computer Science and Engineering, Taibah University, Medina, Saudi Arabia
[3] Computer Science Department, College of Computer and Information Sciences, Imam Mohammad Ibn Saud Islamic University (IMSIU), Riyadh, Saudi Arabia
* These authors contributed equally to this work.



## ABSTRACT

This article introduces the Social-Emotional Nurturing and Skill Enhancement System (SENSES-ASD) as an innovative method for assisting individuals with autism spectrum disorder (ASD). Leveraging deep learning technologies, specifically convolutional neural networks (CNN), our approach promotes facial emotion recognition, enhancing social interactions and communication. The methodology involves the use of the Xception CNN model trained on the FER-2013 dataset. The designed system accepts a variety of media inputs, successfully classifying and predicting seven primary emotional states. Results show that our system achieved a peak accuracy rate of 71% on the training dataset and 66% on the validation dataset. The novelty of our work lies in the intricate combination of deep learning methods specifically tailored for high-functioning autistic adults and the development of a user interface that caters to their unique cognitive and sensory sensitivities. This offers a novel perspective on utilising technological advances for ASD intervention, especially in the domain of emotion recognition.

## INTRODUCTION

In prehistoric times, before the emergence of linguistics and intelligible speech, human communication was predominantly implicit as our distant ancestors relied on signals, facial expressions, body language, and obscure grunting and guttural sounds to convey information as well as to discern the thoughts, feelings, and intentions of others within their community. Today, despite the universal use of spoken language, interpersonal sociality nevertheless remains considerably dependent on nonverbal forms of communication, with experts estimating that close to 80% of all communication is nonverbal and it is deduced that nonverbal exchange was the predominant means of communication and deemed spoken language as its mere extension (*Barrett, Weinshank & Gottleber, 1981*).

Corresponding authors
Adeeb Sait, adeebzeus@gmail.com
Tawfik Al-Hadhrami,
tawfik.al-hadhrami@ntu.ac.uk

Unfortunately, those with autism spectrum disorder (ASD) are not able to understand or produce paralinguistic cues. The Diagnostic and Statistical Manual (DSM-5) for ASD lists deficiencies in understanding and use of gestures, to a complete lack of facial expressions and nonverbal communication, as important criteria for diagnostic assessment of autism (*Autism Speaks, 2013*). People with ASD are known to have difficulties in social interaction (*Pavez et al., 2021*). An individual's quality of life may be negatively impacted by ASD (*Thaler, Albantakis & Schilbach, 2020*). Studies have indicated that a diagnosis of autism spectrum disorder (ASD) is associated with a higher risk of mental health problems. Specifically, social anxiety, generalised anxiety disorders, and depression are common in 60% of individuals with ASD (*Russo-Ponsaran et al., 2016*). As a result, many of these people are prone to loneliness and find it difficult to form healthy social networks. Furthermore, because they struggle to probe social settings effectively, about 78% of individuals with ASD are unemployed, which makes it difficult for them to become financially independent and eventually fully independent.

At present, there is no universally accredited cure for autism, but researchers continually make efforts to provide support and services to individuals with ASD that could help them live a more fulfilled and socially richer life. Some alleviation methods include emotion recognition therapy, which is a manual process that requires direct interaction between the individual with ASD and a behavioural expert. However, this conventional process has proved to be intimidating to patients and is therefore ineffective as a treatment plan (*Russo-Ponsaran et al., 2016*). Social and communicative abnormalities are also less likely to witness any improvement through a traditional intervention process, possibly due to the heavy dependence on a facilitator that may be overstimulating to those with ASD (*Silver & Oakes, 2001*).

Considering the ways that electronic resources might be used as an alternative or in conjunction with normal intervention practises is crucial, especially in light of the failure of traditional methods to alleviate the difficulties faced by individuals with ASD. Because of its freedom from social demands, explicit routines, clear instructions, and consistent responses, as well as its focus on a computer screen that reduces the amount of sensory information fed to the individual, computer intervention appears to be a particularly promising mediation for people with ASD. Self-governance of learning is also made possible by computer programmes, since they allow the user to be more involved in decision-making and pace of learning (*Silver & Oakes, 2001*). There are a number of technologically based intervention tools that have been proposed to help individuals with an ASD diagnosis improve their ability to recognise emotions. Some of these tools centre around the use of augmented reality (AR) and virtual reality (VR) to simulate a social environment and place the individual at its centre in an effort to promote increased social awareness in a controlled setting (*Ko, 2018*). Some focus on computer-aided systems (CAS), like wearable technologies, which may assist people in navigating daily life (*Jaliaawala & Khan, 2020*). Computer-vision-assisted technology (CVAT), which combines deep learning (DL) and machine learning (ML) applications, appears to be the most promising approach for intervention strategies. Deep learning, a prevailing branch of artificial intelligence (AI), is a process that aims to imitate the workings of the human brain

(*Wu et al., 2017*). Deep learning models make use of neural networks, which utilize mathematical algorithms to perform classification and decision-making tasks (*Pavez et al., 2021*). Computer vision employs these algorithms to help guide machines in understanding the contents of digital data such as images and videos. Using computer vision, machines can derive meaningful information from input data to carry out appropriate actions and deliver recommendations that are consistent with the findings.

In the context of emotion recognition, computer vision typically draws on facial expression analysis algorithms (*Kosti et al., 2017*) that aim to interpret nuances in faces to assign an emotional state to the observed actor. Facial expression recognition (FER) algorithms are often based on the Facial Action Coding System (FACS), which consists of a set of anatomically based action units (AUs), individual muscle movements that correspond to specific facial expressions. FER computer vision systems traditionally map facial features (geometrically and/or appearance-based), to AUs, and then use the combination of AUs to determine the corresponding expression or emotion. This article proposes a Social-Emotional Nurturing and Skill Enhancement System for ASD (SENSES-ASD) that is based on facial emotion recognition utilizing deep learning technologies. An ensemble convolutional neural network (CNN) was implemented for a facial expression recognition (FER) that can take images, videos and real-time stream through a web application to predict emotional states based on seven basic classes, namely angry, happy, sad, neutral, surprise, fear and disgust.

The article begins with an introduction that discusses existing solutions for autism diagnosis and includes relevant case studies. The 'Related Work' section introduces the SENSES-ASD web application program and the rationale behind its development. The 'SENSES-ASD System Proposed' section overviews neural networks and deep learning technology and highlights the model architecture and dataset used for the proposed solution. The 'Results and Discussion' section outlines the key findings from the research and implementation of the Facial Expression Recognition (FER) system. It includes a system analysis regarding performance, accuracy, and effectiveness. The article concludes with a 'Conclusion' section that summarizes the main findings and discusses potential future directions for the solution.

## RELATED WORK

### Historical work on ASD and technological interventions

Advancements in digital media technology and the widespread proliferation of electronic devices render digital media platforms fitting as a treatment approach for enhancing emotion recognition in individuals with ASD. Individuals diagnosed with ASD exhibit a strong inclination towards electronic devices with various studies showing that people on the spectrum average longer screen times for electronic media, with ASD severity being directly correlated with screen time (*Dong et al., 2021*).

The affinity for the use of electronic devices in people diagnosed with ASD may be partially owing to the superior "systemizing" ability that is characteristic in such individuals (*Golan et al., 2010*). Systemizing can be defined as the appeal to analyze the variables of a system, in an attempt to determine the rules that regulate its behaviour,

helping one to predict and control its functioning. Individuals with ASD are known for their hyper-attentive nature and their preference for rule-based settings, features fundamental to systemizing. In addition, individuals with ASD have shown enhanced—even superior—detail orientation skills, as well as skills related to regulating and controlling systems—than non-ASD controls (*Baron-Cohen et al., 2003*).

The fascination that individuals with ASD have with computers has led to an increasing amount of study into the potential use of computer programmes and electronic content in ASD intervention services. According to the systemizing theory of autism, people with the diagnosis may use their systemizing aptitudes to make up for their lack of empathy. In light of this, a system of emotions that makes use of technology to categorise expressions into discrete groups might be put into place and used to assist patients in identifying and expressing emotions (*Golan & Baron-Cohen, 2006*). A wide variety of software-based remedies have been created to alleviate expression analysis challenges. A few of these training regimens are reviewed in the parts that follow. They are divided into three categories: interactive programmes, programmes based on CVAT, VR, and AR technologies, and DL and AI interventions.

## Interactive programs

**FaceSay:** FaceSay (*Hopkins et al., 2011*) is an interactive program that aims to teach emotion recognition to children. The program compels ASD patients to take part in activities that address well-known challenges that individuals face when attempting to read faces. As such, the software contains three different games that make use of avatars. The first game "The Amazing Gazing" is an interactive problem-solving task that aims to help the child attend to eye gaze by instructing the user to track the avatar's eye movement whilst completing a task presented by the avatar. The remaining two games "Band Aid Clinic," and "Follow the Leader" attempt to enhance global interpretation skills by directing the user to focus on a collection of facial features, rather than a specific part of the face, to determine the emotion.

The first game is predicated on the knowledge that abnormal reactions to direct stare and avoidance of mutual eye contact are two of the main indicators of an ASD diagnosis. The eyes are invaluable indicators of a person's inner emotional condition since they may express a wide range of social cues. The underlying concept of the remaining games is that individuals with ASD, who display a de-tail-oriented cognitive style rather than an all-encompassing one, lack core coherence, which is defined as "the ability to infer overall meaning from a mass of details." As a result, the last two games aim to teach people with ASD how to observe in a more comprehensive way rather than depending just on isolated traits or expressions.

The findings showed that children with low-functioning autism (LFA), or those who exhibit the most severe signs of the illness, did not significantly improve in their performance. But the findings showed that, compared to their pre-training performance, children on the higher end of the spectrum were significantly better at interpreting emotional states from speech and facial expressions after receiving 10–20 h of emotion recognition training over a period of 10–15 weeks.

**MiXTM:** *Russo-Ponsaran et al. (2016)* developed the MiX program, which is a web-based training tool for micro expression recognition. The purpose of the programme was to improve adult recognition of facial emotions. The computer software integrates educational content with dynamic, authentic faces of folks from various backgrounds. Performance assessments and exercises in facial recognition are also included in the programme. Additionally, video footage is offered to depict seven different emotions: fear, joy, sadness, rage, disgust, and contempt. Two assessors with expertise in emotional decoding assessed the adults' facial expressions in the video. Videos that show the expression of various emotions are displayed on the screen, and the user can manage the video playing by stopping and starting it whenever they choose to encourage paced learning. Results of an experiment using the MiX program showed considerable improvement in emotional awareness in children, as well as an enhanced ability to self-express internal emotional states.

## CVAT, AR, and VR interventions

VR provides a system by which real-world scenarios can be simulated using CVAT and computer graphics, providing a repeatable but equally diverse environment for learning. The ability to control stimulus using AI, whilst also inculcating human-like senses is suitable for ASD intervention and has shown potential for learning and evaluation.

*Serret et al. (2014)* developed a program named "JeStimule", a CAS which matches facial expressions with colours. In this system, each of the seven basic emotions is associated with a colour (*e.g.*, happy = yellow, anger = red, *etc.*). The system's target audience is individuals with low-functioning autism, apparent in its lack of use of textual information and its vehement dependence on the assigned colour codes. The intervention required subjects to determine the emotional expression of a character, as well as predict what expression would be produced in relation to a certain trigger. No conclusion was reached regarding whether or not emotion recognition was improved.

*Strickland, Coles & Southern (2013)* developed a tool to aid individuals with mild ASD or High-Functioning Autism (HFA), who are capable of functioning with a certain degree of independence but may still encounter challenges in social-emotional functioning.

The "JobTIPS" tool was designed to assist participants in acquiring abilities relevant to the workplace, such as those needed for a job interview. It was based on the CVAT. The software was built on virtual learning environments, visual aids, and tutorials. Twenty-two young individuals with mild ASD or HFA participated in an experiment designed by the researchers to assess the usefulness of the CVAT. Two groups were created from the group: one for the intervention and the other for control testing. The participants who had undergone the intervention had improved verbal abilities in comparison to the control group during a simulated job interview, according to the findings.

*Cheng et al. (2005)* employed a VR avatar as a CVAT tool for intervention. The avatar was designed to aid emotional recognition in individuals with ASD. Facial expressions and animated sequences that conveyed emotion were enacted by the avatar, with expressions of happiness, sadness, fear, and anger. The study was conducted in three phases. In the initial stage, participants were tasked with identifying the facial expressions produced by the

avatar. The second phase required users to predict the emotional state from the facial expressions identified. The objective of the third phase was to improve the ability of users to predict situations that may elicit those particular emotional responses. The author reported that 90% of the participants demonstrated proficiency in discerning, forecasting, and interpreting the emotional expressions acted out by the avatar.

Another study (*Fabri, Elzouki & Moore, 2007*) avatar interaction using training stages similar to those used by *Cheng et al. (2005)*. In the trial's first stage, 34 children with ASD whose ages ranged from 9.96 to 9.96 on average were asked to choose the best description for the avatar's expression. In the second round, participants had to guess the emotions that a certain event would evoke while watching the avatar act it out. In the third phase, the participants were required to pinpoint the exact reason behind the emergence of a specific emotional condition. Of the 34 participants, thirty successfully finished all three phases of the intervention the four remaining participants exhibited symptoms that coincide with level 3 autism and therefore faced greater difficulty in completing the tasks so this work will target such issues.

## Deep learning interventions

Several computer-based technologies have been developed that aim to enhance the ability to detect human emotions and attitudes. These technologies typically utilize cameras built into computers or mobile phones in order to capture still images and frames of faces, that would subsequently be analyzed to determine emotional state. Most facial recognition deep learning models are generalized and fail to implement the technology as an intervention method for ASD. Some models like the one in *Talaat (2023)* are implemented in a way opposing to the one proposed in this research, by instead using the model to detect the emotion of those diagnosed with autism, owing to the deficits in self-expression that autistic people exhibit.

In a recent publication (*Dhuheir et al., 2021*), a novel healthcare systems were reviewed, which employs a deep neural network (DNN) for assessing patients' mental health conditions. The system employs real-time image capture using cameras installed in healthcare facilities. The normalization method was employed by the authors to pre-process the input data, to enable the identification of facial features present in the image and to determine any relevant points. Following this, feature extraction was carried out and feature selection was performed based on two datasets, the Extended Cohn-Kanada (CK+) and the Japanese Female Facial expression (JAFFE) datasets. Ultimately, the process of emotional classification was performed with a SoftMax classifier, which effectively categorized each input image into one of six distinct emotional categories. The proposed system also included speech emotion recognition capability. *Li & Wang (2018)* proposed the use of a facial recognition deep learning model in an e-learning system. The approach is based on the implementation of a deep neural network and support vector machine. Its main objective is to detect and examine the facial expressions of students in real-time during online learning, in order to construct a model of their emotional responses and determine engagement levels. Facial recognition technology is employed using the

OpenCV and Dlib python libraries to analyze the captured facial features of students and enable the implementation of a global feature recognition method.

*Talaat (2023)* suggested an Internet of Things (IoT) smartphone application for real-time emotion recognition system that helps carers of autistic children understand their feelings. A wide spectrum of emotions, such as neutrality, fear, joy, sadness, and surprise, can be detected by the suggested system. The Cloud Layer, Fog Layer, and IoT Layer are the three separate layers that make up the CNN used in the study. The way this application works is that it first uses the camera on the phone to take a picture of the youngster, which it then sends to the fog layer. The fog layer uses a controller called a fog server, which uses a pre-trained model that is stored within it to do the task of face expression identification. The fog layer also encompasses the database of images that were used for training the model. Upon successful emotion detection, the fog server transmits an alert to the caregiver's device in the event that a negative emotional state is detected.

## Limitations with existing solutions

The thorough literature review shown above indicates that CAS interventions have a lot of potential. Despite the fact that the majority of these studies show notable progress, none provided clear proof that individuals with ASD can attain full emotional recognition. Most research conducted limited testing, and implementation was performed in a therapeutic/clinical way and without many capabilities for implementation outside a controlled environment. The implication is that autistic individuals who are observed to have issues with generalization (*de Marchena, Eigsti & Yerys, 2015*), that is, the adaptation of a particular strategy across different but comparable contexts, may do well within the controlled setting but fail to apply what they have learned in real-life scenarios. Moreover, many of the existing intervention methods are catered towards children with ASD and fail to consider solutions for adults who were diagnosed with autism during later life.

Since the researchers' solutions are intended to assist children with the disorder, they also believe that employing a variety of visual stimuli, such as vivid colours, effects, and images (as in the FaceSay programme), will aid in the child's ability to concentrate on the current activity. While this could help neurotypical kids pay attention, more sensory input might cause overstimulation, behavioural issues, and a loss of interest in learning in kids with ASD (*Gaines et al., 2014*). Furthermore, current research describes the CAS-based intervention methods in remedial ways, discussing any observations and findings but failing to go into depth on the architecture and design of the employed system. The choice of structure and utilized algorithms within the CAS interventions are therefore rarely reviewed or justified (*Jaliaawala & Khan, 2020*).

## THE SENSES-ASD SYSTEM PROPOSED

The SENSES-ASD system proposed in this article represents a novel approach to ASD emotion recognition training, employing cutting-edge deep learning methodologies for emotion recognition. This system is a blend of a user-friendly web application interface and a robust deep-learning model. The web application acts as the interaction platform for

users, while the deep learning model, the heart of the system, performs facial emotion recognition (FER).

## Algorithm for the SENSES-ASD system

1) **Input:** The user interface of the system accepts input data (images, videos, or real-time streams) from the user.
2) **Pre-processing:** This input data undergoes pre-processing to prepare it for emotional recognition, which may include face detection, normalization, resizing, and grey scaling.
3) **Emotion recognition:** The pre-processed data is fed into the deep learning model, specifically an ensemble convolutional neural network (CNN), for emotion recognition. This model is a scaled-down version of the Xception CNN model, trained on the FER-2013 dataset.
4) **Output:** The model predicts the emotional state, classifying it into one of seven basic classes: angry, happy, sad, neutral, surprised, fearful, and disgusted.
5) **Feedback:** The web application receives the model's prediction and provides real-time feedback to the user.
6) **Repeat:** The algorithm then repeats from step 1, facilitating continuous real-time emotion recognition.

This innovative solution offers a marked advantage over traditional emotion recognition systems, as it possesses the capability to contextually analyze facial expressions. This allows for a more comprehensive assessment of emotions and ensures a more realistic training experience, thereby increasing the probability of generalizing these learned skills to real-world scenarios.

While many existing ASD solutions predominantly target children, the SENSES-ASD system is designed primarily for adults over the age of 18 diagnosed with level 1 (high-functioning) autism. This specific target group was chosen based on research indicating their potential for substantial improvement in emotion recognition following the use of traditional computer-aided systems. By focusing on this group, an important benchmark can be established, offering insights into the unique emotion recognition challenges faced by the ASD community. This could, in turn, pave the way for the exploration of advanced computer-based intervention strategies designed to address the intricate interpersonal struggles experienced by individuals across the entire autism spectrum.

With the integration of deep learning into the domain of ASD interventions, the SENSES-ASD system heralds a new era, aiming to redefine the existing standards in ASD intervention strategies. In future, the plan is to enhance the system's capabilities further, refining its accuracy and broadening its application to cater to a wider range of users within the autism spectrum.

The SENSES-ASD system, developed on the Flask web framework, integrates a tailored convolutional neural network (CNN) designed for nuanced emotion recognition. Unlike general-purpose models, this CNN specifically addresses subtle facial expressions, often characteristic of individuals with ASD. The model incorporates spatial hierarchies and

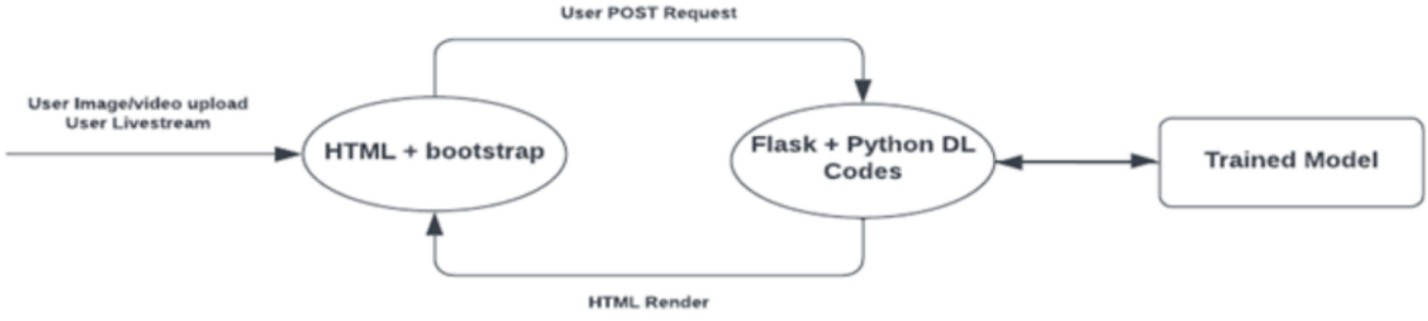

**Figure 1 Web application structure.**

attention mechanisms, enabling it to detect minute facial cues. To enhance its robustness, ensemble techniques are employed, combining the decision-making of multiple models for a more balanced output. Furthermore, the system features dynamic learning, allowing it to adapt and refine its recognition capabilities based on user interaction.

## Web application structure

As illustrated in Fig. 1, HTML and JavaScript were used to support the front-end development of the FER software, while the deep learning code used for emotion recognition was integrated within the Flask back-end framework.

When a user issues a post request by either uploading an image or video or requesting real-time streaming, a POST request is initialized to send the file or live capture to the backend for processing, which then loads the locally saved trained model to determine the emotional expression.

The web application is comprised of three modules (image upload, video upload, real-time streaming) that can be used for facial emotion classification based on the seven basic emotional states of happiness, sadness, surprise, fear, disgust, anger, and neutrality. The functionalities of each module are highlighted in the following sections.

## Image upload method

The first module is a web page where users can upload an image of a face that may be displaying a particular expression. Once the image is processed and passed through the CNN, the emotion is determined and declared aloud. The use of speech generation to read digital data was deemed appropriate as research has shown that decoding and word apprehension abilities are improved with text-to-speech (*Stodden et al., 2012*). This is especially advantageous as literature is increasingly highlighting deficits in textual comprehension for people on the lower end of the autism spectrum (*Gately, 2008*).

The web application shows the image upload functionality. In this case, an image file that captures an image of an angry man can be uploaded, for which the FER application successfully generates an audible description of the emotion.

Initially, the user submits an image file which is saved to a directory on the server. The image file is then read by the underlying system which proceeds to perform face detection

using a pre-trained face detection model. If a face is found an emotion detection model is applied to predict the emotion label. The predicted label is then converted to speech using a text-to-speech library and played as an audio file. Once the audio file is played, it is removed from the server, and the web application is refreshed to prepare for more input.

## Video upload module

The second module is a page dedicated to deciphering facial expressions from an uploaded video file. It was primarily developed for clinical support and training purposes. The training process using this module would entail capturing a video of a clinician's face with audio from the patient's perspective as the conversation evolves naturally during the training session. The video is then uploaded to the system and analyzed for nuances in facial expressions that are evident during the conversation. By capturing a conversation that is natural and unforced, with both audio and visual components, the patient can be trained on visual information that is more practical, relevant, and realistic, making training more effective. This approach differs from traditional facial expression training videos, which typically use visuals that are irrelevant to the patient and capture exaggerated emotional expressions that lack authenticity and do not accurately depict real-world responses to events.

Pattern recognition involves interpreting a visual pattern in a global, cohesive manner and being able to predict what the next pattern could be in a sequence of methodical visuals. Individuals with autism tend to show heightened pattern recognition abilities (*Crespi, 2021*) as a consequence of their "systemizing brains". By essentially uploading the captured video and "rewatching" a natural conversation they have had unfolding and annotating with emotional labels, individuals with ASD can be trained to recognize and contextualize emotional expressions in social situations. With repeated exposure to such training, patients may develop pattern recognition skills that facilitate the generalization of facial expression recognition and contextualization beyond the initial training environment. This may enable patients to interpret emotional states more accurately and even predict emotional responses based on contextual cues.

As opposed to the image upload component, the video upload module does not make use of text-to-speech functionality on account of the rapid expression variations that are emblematic of natural conversation, and if narrated aloud persistently may be overwhelming and as a result, obstruct the learning process.

In place of speech generation, a boundary encapsulating the face is drawn on the uploaded video with a label that describes the emotional expression shown in the frame. Moreover, another window unveils the occurrence probabilities, out of 100 as a horizontal bar chart, of the seven emotional states for each frame as the video progresses. Initially, only the video with the border and emotional label was developed, but after research (*Anthony et al., 2013*) was reviewed, it was revealed that a large percentage of individuals with HFA have an affinity for numbers. Adding the probability scale may therefore motivate some of the patients during training by indulging their systemizing abilities and appealing to their interests.

## Realtime streaming module

The last module of the web application is the emotion recognition of a live stream video captured by the device's webcam. This component was developed with several applications in mind. Individuals with ASD could use the web application and its real-time emotion recognition capabilities in learning to produce adapted facial expressions with the use of visual feedback (*i.e.*, the emotional labels) in real-time, as the user themselves acts out a particular emotion. This training phase would be conducted after contextual information and corresponding appropriate expressions are acquired from studying videos captured during training sessions with a professional (refer to the previous section). The ASD patients can apply the knowledge they have learned about suitable facial expressions during various topics of conversation practised with their clinician in constructing their own facial responses that better match the emotional content of the conversation. Ultimately, this type of training can help individuals with ASD become more active participants in conversations, improve their social communication skills and foster more effective communication with others.

Another potential application of the real-time emotion recognition system is to incorporate it into a wearable device, such as smart glasses, to interpret the emotional states of others in real time and provide feedback to the user through a connected earpiece. For individuals with ASD, this technology could provide a valuable tool for avoiding potentially hostile interactions by identifying people who may be labelled as "angry" or displaying other negative emotions. This is particularly important given that many people are unfamiliar with ASD and may not respond positively when the individual with the disorder does not adhere to conventional societal norms of communication. By providing real-time information on the emotional states of those around them, individuals with ASD can adjust their behaviour and speech accordingly to reduce the risk of negative encounters.

Additionally, the real-time wearable system could be used in critical situations such as job interviews, where the user could adjust their speech and actions based on the facial expressions of the recruiter. This aptitude for understanding and responding to emotional cues could be crucial in leaving a favourable impression on employers, which in turn raises the likelihood of being hired for the position.

Note that if this functionality were to be implemented in a wearable device such as smart glasses, text-to-speech functionality similar to the one introduced in the image upload module of the software would be administered so that the user could be notified of emotional states through an earpiece. Moreover, the software would need to be modified based on the individual capabilities and shortcomings of each user. For example, a user who struggles with identifying angry individuals or distinguishing a fearful expression from a surprised one, can be notified only of those emotions, reducing the volume of information to be percolated through the earpiece to be absorbed by the user, consequently reducing the chance of overstimulation.

Figure 2 depicts the entire web application workflow which integrates the various modules, with a face detection module and a trained emotion recognition module to make

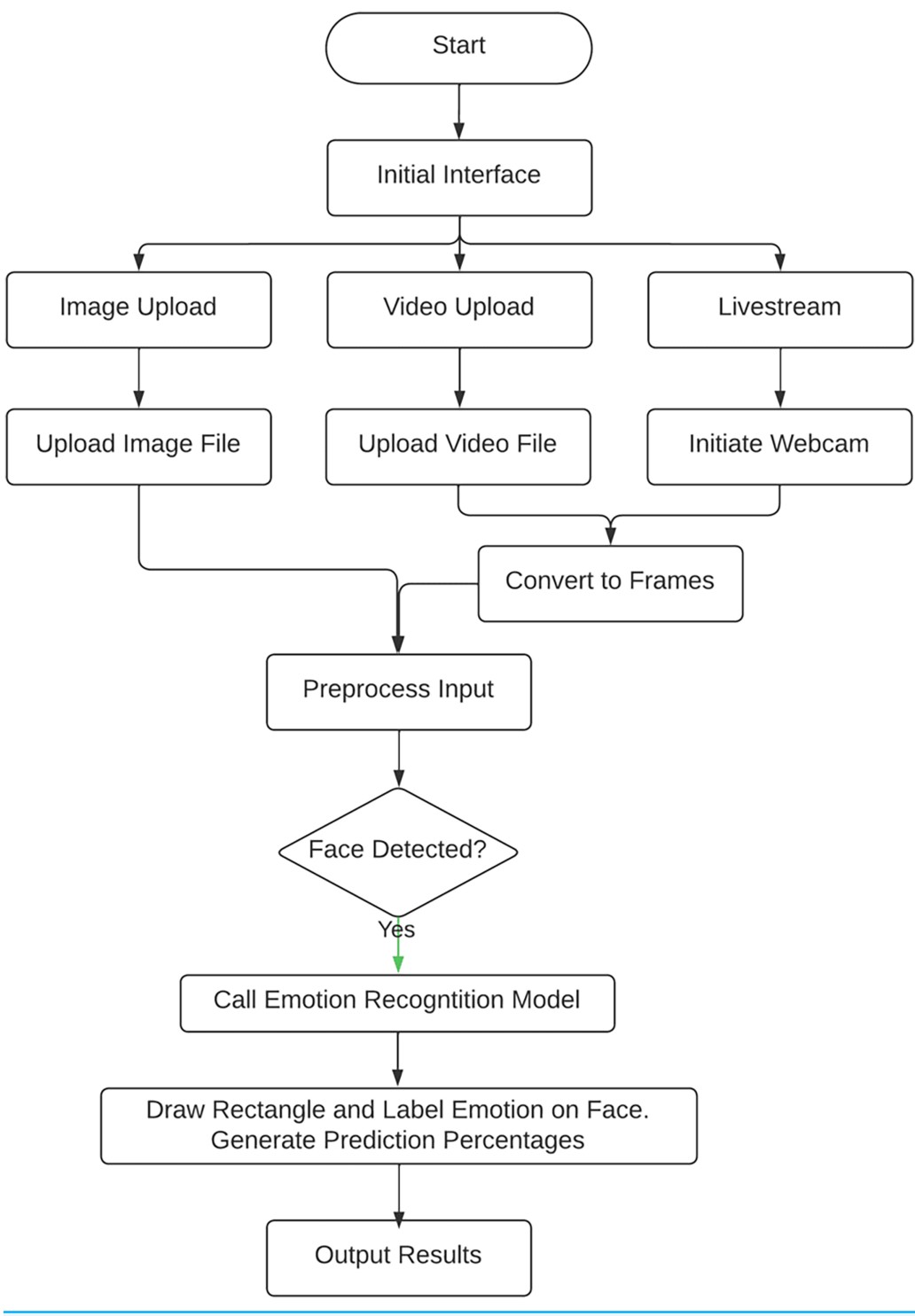

**Figure 2** FER system workflow.

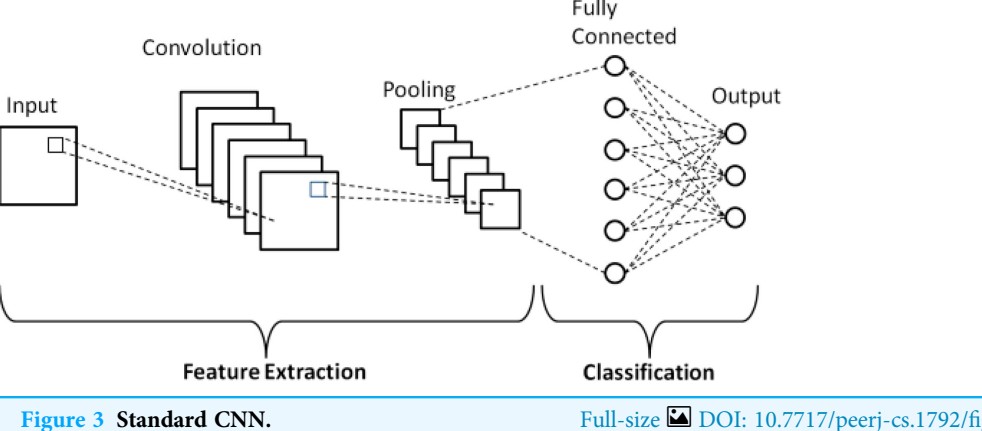

**Figure 3 Standard CNN.**     

emotional classifications. Initially, the user selects to either upload an image file, video file or live stream with their device's webcam. If they choose to capture feed from their webcam or upload a video, it is converted into frames, and the resultant image is pre-processed by resizing it and converting it to grayscale. The image is then fed into the face detection module. If a face is detected, the image or frame is fed into the pre-trained emotion recognition module, which then predicts the emotional state shown and outputs the results to the user (*McNair, 2018*).

## Convolutional neural network (CNN) for SENSES-ASD

The emotion detection model adopted for this solution employs the convolutional neural network (CNN). CNNs, introduced in *Le et al. (1998)*, are a class of biologically inspired, artificial neural networks designed to learn spatial hierarchies of features present in data characterized by a grid pattern (*i.e.*, images). CNNs accomplish this by using multiple building blocks comprised of layers, which perform complex computations with the purpose of extracting features and patterns from the set of input data.

## Standard CNN

Figure 3 shows a standard CNN. As illustrated in the figure there are three main layers typically stacked together to configure the network. The first portion, consisting of convolution and pooling layers is responsible for feature extraction and the next portion, made up of fully connected layers, handles image classification based on the extracted features.

  The following sections outline operations performed by each of the three layers.

### *Convolutional layer*

Convolution, a type of linear operation, is performed in this layer. A kernel/filter is slid across the input image typically consisting of an array of numbers known as a tensor. At each location, the values in the kernel are elementwise multiplied with the corresponding values in the overlapping region of the input tensor. The results are then summed up to produce a single output value for that position in the output tensor, also known as a feature map. Multiple kernels are applied to the input tensor giving rise to several feature maps.

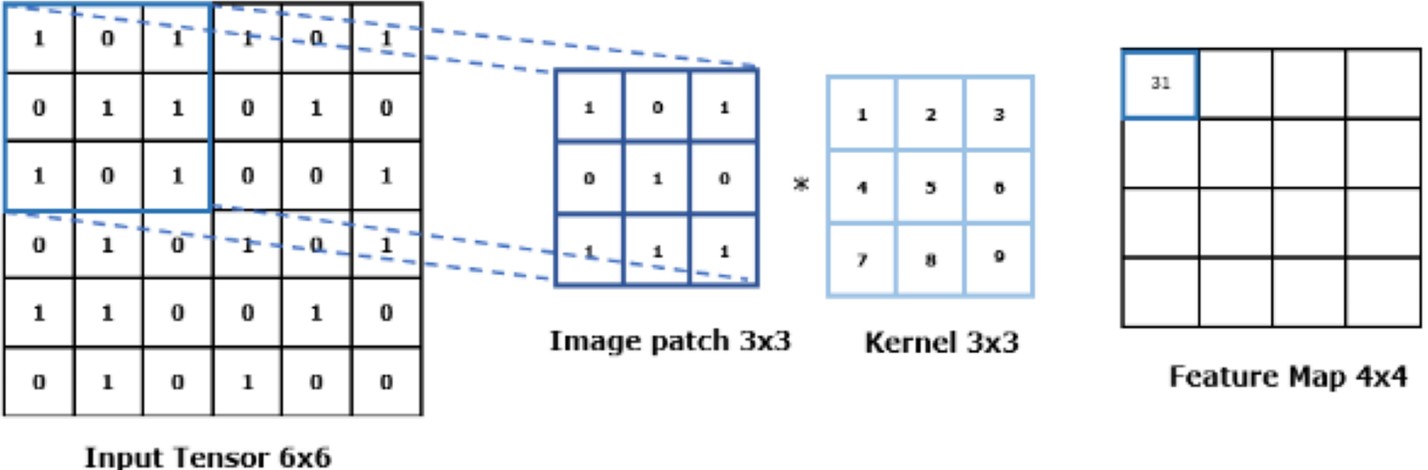

**Figure 4  Feature extraction with CNN.**                     

Given that feature maps capture the presence or absence of features in the input data, kernels can be described as "feature extractors" (*Yamashita et al., 2018*).

The convolution process is defined with two crucial hyperparameters; the size of the kernels and their number. The size of the kernels must be smaller than the input tensor to enable capturing fine-grained patterns in the data, with typical values at $3 \times 3$, $5 \times 5$, and $7 \times 7$. The number of kernels used within the CNN is often arbitrary and depends on the specific requirements of the system. However, an insufficient number of kernels may result in an incomprehensive representation of the input data, whilst too many kernels may make the model more difficult to train and could lead to overfitting, meaning the model performs well on the training data but poorly on any new data.

Figure 4 illustrates the feature extraction mechanism using a kernel in convolutional layers.

### Pooling layer

A pooling layer, commonly configured after a convolutional layer, provides down-sampling operations to the input feature maps produced during convolution. Pooling reduces the dimensionality of the input, more specifically the width and height (whilst maintaining the same depth), keeping only the important features and consequently reducing the number of features to be learned by the model. The pooling process reduces computational needs and resource requirements, culminating in more efficient model training and deployment. Pooling also helps to resolve the problem of overfitting by enabling the network to integrate all image dimensions, introducing a translation invariance that allows the model to successfully recognize the input feature even if it is slightly shifted or distorted (*Ajit, Acharya & Samanta, 2020*).

There are several pooling approaches, including max pooling, average pooling, and stochastic pooling. Because it has been empirically shown to preserve essential features more reliably than other variations, max pooling is the most popular operation.

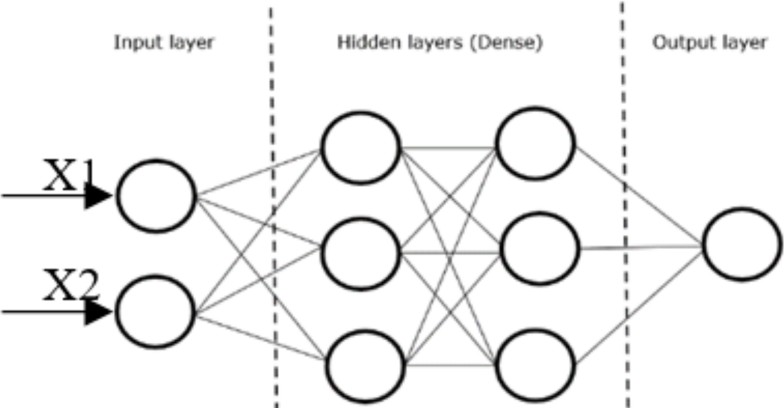

**Figure 5  Dense layer CNN.**

### Fully connected layer

Once the features extracted by the convolution layers and down-sampled by the pooling layers are generated, they are mapped to the final outputs of the network by a set of fully connected layers, collectively known as a dense layer, which ultimately performs the classification task. The feature maps output by the previous layers are flattened—modified to be one-dimensional vectors—with each element representing a feature to be input into the dense layer.

Within the dense layer, each node in a fully connected layer is connected to every node in the previous layer by a set of weights, which are essentially learnable parameters that the network uses to adjust the error between the predicted value and the actual value. These weights are used to determine how the input vectors to that node are transformed to produce its output.

The final fully connected layer usually has the same number of output nodes as the number of classes in the model and produces as an output a vector of probabilities with a score for each possible class.

Figure 5 illustrates the typical layout of a dense layer with two hidden layers. X1 and X2 represent the input vectors to the dense layer, while the lines between the nodes represent the weights that will be used to determine the output of one node, and hence the input of the next.

### Depth-wise convolution

Depth-wise convolution is a spatial convolution performed independently over each channel of an input, followed by a pointwise convolution. This means that rather than applying one filter to the entire input tensor to produce the feature map, a separate filter is applied to each channel to produce a set of intermediate feature maps that represent certain characteristics of the input tensor. The following pointwise convolution combines all the intermediaries to produce the final output feature map.

As shown in Fig. 6, standard convolution applies just one filter to the entire input feature map to obtain the dot product, making the number of parameters learned by the model directly proportional to the dot product of the number of input channels and the

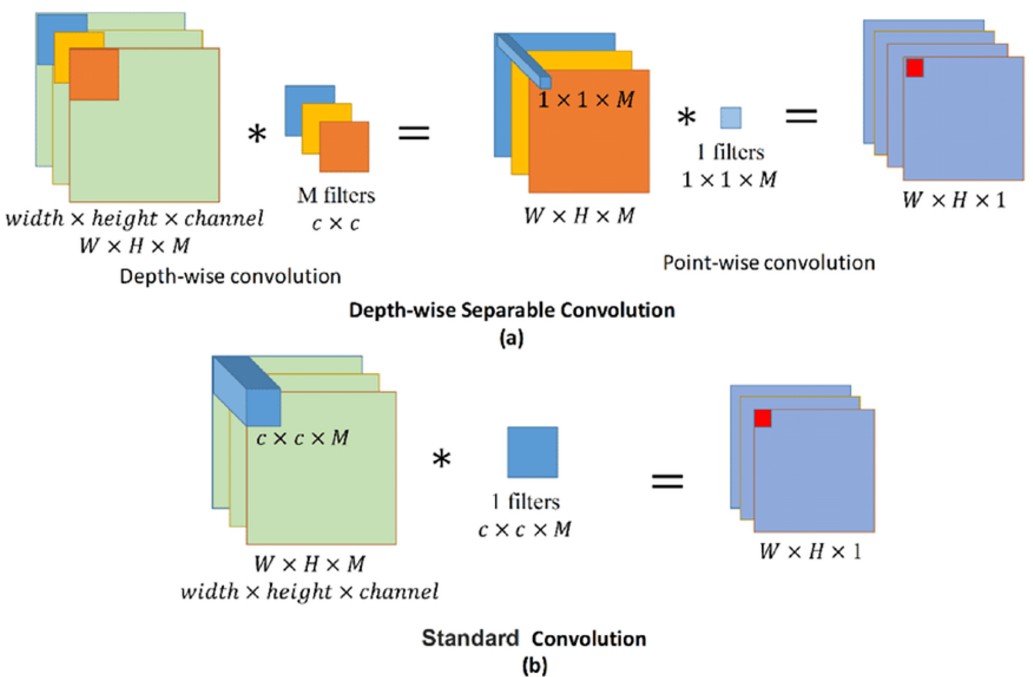

**Figure 6 Standard and depth-wise separable convolutions.**

size of the filter used. In contrast, depth-wise convolution applies a different filter for each of the channels in the input feature map. This makes the number of parameters (or weights) used in depth-wise convolution directly proportional to the number of input channels. The application of pointwise convolution, which uses $1 \times 1$ filters, after the initial depth-wise convolution in the modified setup, means that every spatial location has the same set of weights, and the number of output channels is independent of the number of input channels.

Thus, by breaking down the convolution operation into two distinct steps, similar accuracy to the standard methods can be achieved but with a smaller number of attributes. This further helps achieve the system objective of straightforward training. The combination of depth-wise convolution and removal of the fully connected layers may also enable deployment of the system on devices with limited processing power, helping create a more portable system, if for example implemented in a mobile application or within wearable smart glasses.

## Xception model architecture

The Xception architecture, introduced in 2016 by *Chollet (2017)*, is based entirely on the concept of depth-wise separable convolutions. The model consists of a sequence of depth-wise separable convolution layers with batch normalization and ReLU activation. It does not use fully connected layers and instead has a block with one node and a SoftMax classifier to perform the final classification.

Since the Xception model replaces the standard convolutional layers in the CNN with the more efficient depth-wise convolution operation, it produces a smaller number of parameters and enables faster training times compared to other architectures. The application of batch normalization and global average pooling within the Xception model also reduces the risk of overfitting the training data, yielding a more effective model. The relatively lower computational cost of the Xception architecture also makes it suitable for everyday use by ASD patients who may not have access to more powerful devices, increasing product accessibility and allowing a greater number of individuals to improve sentiment analysis capabilities. This is driven by an effort to meet the aim of creating assistive technology that is more inclusive and more readily available to individuals diagnosed with the disorder, at the minimal possible cost.

## The mini Xception model

The mini Xception architecture is a simplified, more compact version of the Xception model, with a reduced number of layers and weights. This modified architecture maintains the primary constituents of the original model, including separable convolution, batch normalization, and residual networks, but with reduced complexity.

The lower complexity of the mini Xception characterizes it with a lightweight nature that reduces computational needs, making a FER system implementing this model suitable for use within mobile devices and other embedded systems.

The architecture starts with two consecutive convolutional layers with eight filters, each with a subsequent batch normalization operation and a ReLU activation.

Following the initial convolutional layers are four convolutional blocks each consisting of a residual connection made up of a convolutional and batch normalization layer. Each block is also comprised of two separable convolutional layers with ReLU activation and batch normalization, succeeded by a max-pooling operation for down sampling. With each consecutive separable convolution block, the filters gradually increased, starting with 16 filters in the first block, and ending with 128 filters in the fourth.

The output of the down sampling and residual connection are combined using element-wise addition, to be passed into a convolutional layer, global average pooling operation, and ultimately SoftMax activation to generate a definitive occurrence probability for each class of emotion ('angry', 'happy', 'sad', 'neutral', 'surprise', 'fear', 'disgust'). This architecture (Fig. 7) would produce approximately 60,000 parameters, a number significantly lower than that of other architectures for the same accuracy, and at a reduced computational cost.

## Dataset

The accuracy and effectiveness of an emotion detection model are highly reliant on the dataset of images used to train the model (*Goodfellow et al., 2013*; *Lucey et al., 2010*; *Real-world Affective Faces Database, 2023*; *Behera et al., 2021*). For example, the FER-2013 proposed in *Goodfellow et al. (2013)* has 35,887 greyscale images measuring 48 × 48 each and classified with labels based on seven emotion classes; anger, disgust, happiness, sadness, surprise, fear, and neutral. The images are split into two subsets, validation data

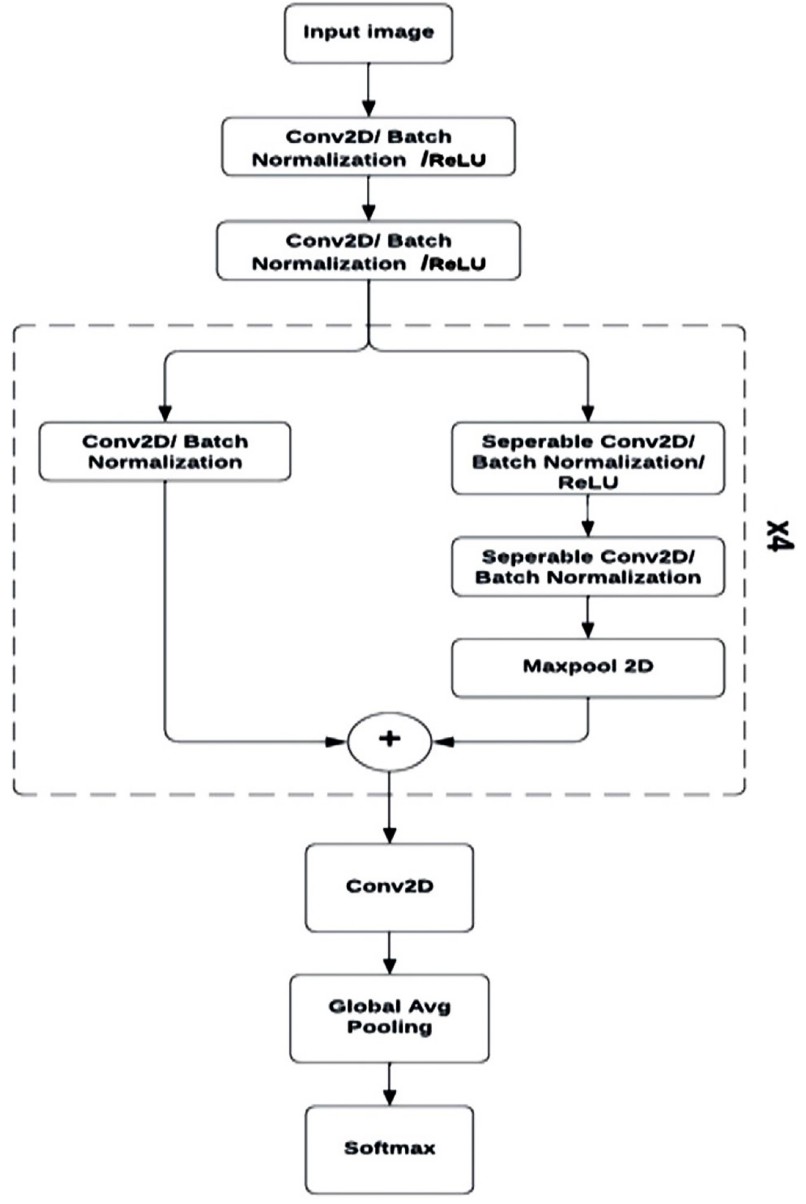

**Figure 7 Mini Xception architecture.**

used to validate the performance of the FER model during training, and the actual training data used to train the model. The FER2013 does not have any participants and instead is formed from a collection of images obtained from a variety of different sources, including Google searches and self-surveys.

The FER-2013 was the suitable choice for the emotion recognition model, its large size, diverse emotional categories, and resource pools contribute to robust model training. Moreover, the dataset is standardized and is essentially an established benchmark for emotion detection models, using it, therefore, helps achieve the standardization objective that is imperative in developing potent emotional training regimes for individuals with ASD.

All networks are to be implemented using the Keras framework and trained on the NVIDIA MX130 GPU.

It is proposed that a batch size of 32 is used so that each iteration processes 32 images sequentially rather than the entire dataset all at once. This number was deduced optimal as any larger batch size would drain GPU memory and lead to sub-standard model generalization, especially considering the low computational power of the used GPU. The use of a smaller batch size can therefore speed up training whilst also increasing generalization. The number of epochs, meaning the number of times the entire FER-2013 dataset is used during the training process, is set to 100, this number is proposed to achieve sufficient model accuracy without causing the model to become specialized on the training data.

During training, after each iteration/batch of data, a mathematical function known as a loss function is generated. The loss function is used to measure the difference between the predicted output of the emotion and the actual output, providing a quantitative measure of the performance of the model by measuring the error/cost of the model's predictions. Within the solution proposed in this article, the use of the 'categorical cross entropy' function is appropriate. This loss function is standard and has been proven effective particularly for multi-class classification tasks, making it suitable for classification of the seven emotional states described in this solution.

To minimize the loss function, the Adam optimizer, which uses a variation of the stochastic gradient descent (SGD), is suggested. Optimization using gradient descent in model training is used to adjust model weights/parameters with the purpose of minimizing the loss function and therefore the cost of the model's prediction, this is done by finding the minimum of the function by iteratively moving in the direction of the steepest decrease of the function. The choice of the Adam optimizer was mainly due to it being able to strike a good balance between the speed of training and its accuracy. The Adam optimizer uses a combination of two techniques, namely momentum and adaptive learning rates, to accelerate the convergence of the training process and improve the generalization performance of the model, whilst adaptively adjusting the learning rate—the rate at which the optimizer adjusts the model's parameters during training—of each weight based on the magnitude and direction of the gradient. It also requires less memory to store the model weights and gradients compared to other optimizers, making the optimization process more computationally efficient (*Arora et al., 2021*).

Other techniques proposed for use in the training of the FER model include data augmentation to increase the diversity of training data by transforming the training images (rotating, shifting, flipping, *etc.*). Moreover, early stopping of the training process, and learning rate reduction which reduces the rate by a certain factor if validation loss does not improve after a specific number or epochs, are proposed for use within the model to reduce the risk of overfitting. Checkpointing should be used to save the best model obtained from training based on the validation loss. Finally, the model performance is to be evaluated using a test set of images, and the accuracy and loss of the model should be recorded and visualized for future reference.

**Table 1 Comparison of trained models.**

| Model employed | Number of epochs | Validation accuracy % |
|---|---|---|
| Simple CNN | 34 | 49 |
| Tiny Xception | 52 | 54 |
| Mini Xception | 102 | 60 |
| Big Xception | 88 | 58 |

The model training process constitutes feeding in the FER-2013 training dataset, the model then attempts to learn emotional classifications of the images by comparing its predicted output label with the original label designated for each image.

After completing the training process, the model's performance is assessed based on the FER-2013 validation set, citing several metrics that include validation accuracy as well as validation loss. The model is also saved locally and can then take in test images of varying sizes and colours, pre-process them, and predict emotional labels of faces shown in the images.

## Model exploration and training

This process entails feeding the training algorithm with the training dataset. The algorithm will process the images and generate a model that is capable of deducing emotional states from a new unseen image or frame. A supervised learning method was appropriate since the FER-2013 dataset is prelabelled with emotional classes. During the supervised learning process, the algorithm essentially learns a function that maps the input emotional variable to an output variable by being trained on the labelled dataset where the output values are already known.

Numerous models were trained to determine the model that produces the most accurate emotion classifications. Initially, a simple network was trained to establish a performance baseline, the baseline metric was then used to provide a comparison point for the effectiveness of later models. Following the simple CNN training, training was conducted on the Tiny Xception, mini Xception, and Big Xception neural networks.

Table 1 summarizes the performance metrics of the various models used. As shown in the table, the simple CNN achieved the lowest accuracy of 54% in 34 epochs, after which the performance on the validation set failed to improve. This was taken as the reference point to evaluate the other models. The most effective model was the mini Xception model with a validation accuracy of 60% in 102 epochs, and as such was employed for use in the FER software.

## Model refinement

To refine the performance of the mini Xception model, iterative changes were made to the model's architecture, hyperparameters and the training process. Initially, the default hyperparameters set by Keras were used and produced a validation accuracy of 60%. Various hyperparameters were adjusted after the initial training to optimize the model and improve performance.

Initially, the learning rate was fine-tuned using values that range between 0.0001 and 0.01. It was found that a learning rate of 0.001 was optimal and produced the best validation accuracy of 62%. Experimentation with the batch size was also performed with sizes ranging between 16 to 64. It was deduced that although it did not significantly increase the model's accuracy, a batch size of 32 was suitable since it did not diminish the model's accuracy.

Regularization techniques were applied to the model to reduce the risk of overfitting. Dropout layers with a drop rate of 0.5 were added after each convolutional block. Adding the dropout layers increased the validation accuracy to 64%.

During training, the loss and validation accuracy were monitored on the training set as well as the test set to identify any bottlenecks and performance issues. It was discovered that after around 80 epochs, the validation accuracy was subsiding, it was inferred that the model had overfit to the training data and was failing to generalize well to new data. Consequently, an early stopping call-back was added to halt the training process if the model accuracy failed to improve for five consecutive epochs. This improved the final accuracy to 66%.

Following model training on the FER-2013 dataset, an in-depth analysis of the model was conducted. Matplotlib was used to provide visualizations of the model's performance across different evaluation criteria. Mainly, model loss and accuracy during both training and validation. A confusion matrix was also generated to assess the model's performance by comparing the predictions made for each emotional class to the actual labels used in the dataset.

Whilst no direct testing on individuals with ASD was conducted, a qualitative measure of the suitability of the software as an intervention was inferred based on analysis of the quantitative results.

## RESULTS AND DISCUSSION

### Loss function analysis

The loss function, utilized to quantify the divergence between the predicted emotional labels and ground-truth labels, commenced at approximately 1.80 for both the training and validation sets. Throughout 80 epochs, this value subsided to around 0.80, manifesting the model's increasingly precise classifications.

However, as indicated in the graph (Fig. 8), episodic surges in validation loss were observed, suggesting a tendency towards overfitting. This implies that the model may not generalize well to new, unseen data, possibly due to disparities between the training and validation sets.

### Accuracy metrics

Initially, the model's accuracy was around 20% for the training dataset. Through progressive training cycles (epochs), the accuracy escalated to about 71%.

Conversely, the highest validation accuracy recorded was 66%, a discrepancy of 5% when compared to the training accuracy, further corroborating the overfitting hypothesis as shown in Fig. 9.

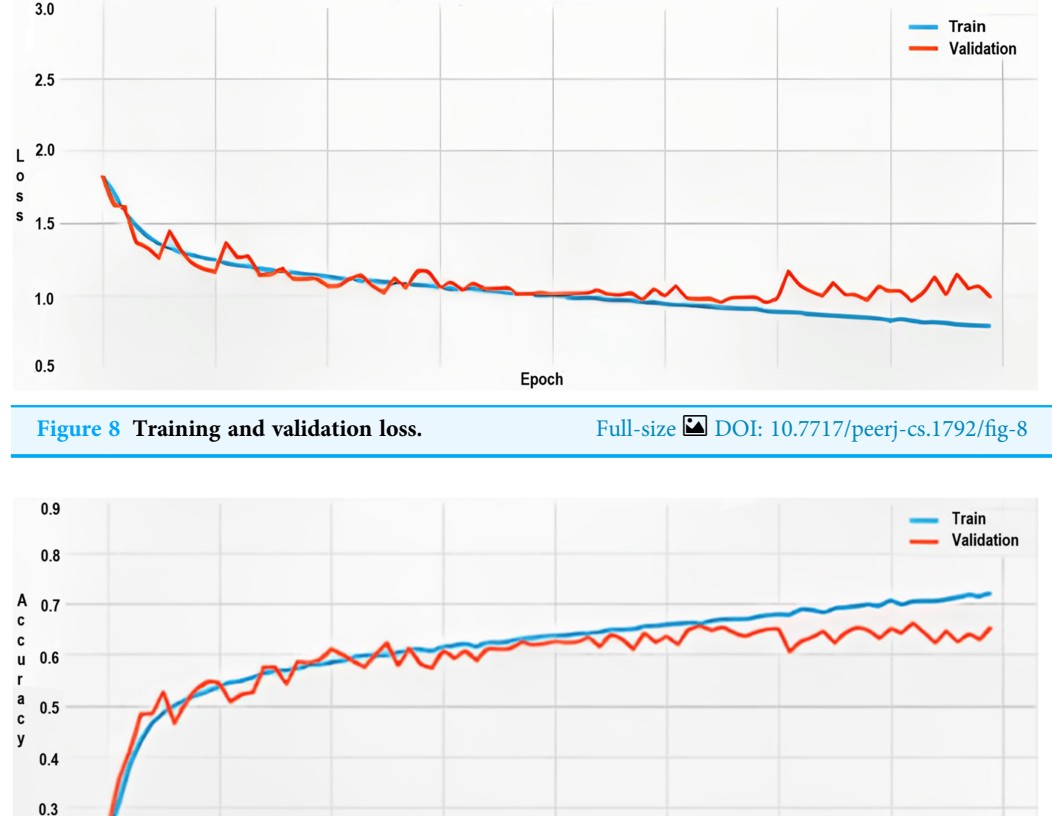

**Figure 8  Training and validation loss.**   

**Figure 9  Training and validation accuracy.**   

## Confusion matrix analysis

The confusion matrix provided a comprehensive view of the performance of the facial emotion recognition system. The matrix visualizes the number of images that were correctly classified by the model for each state, as well as the number of images for each class that were misclassified as other emotions.

The confusion matrix as shown in Fig. 10 illustrated that the model was most proficient at identifying 'happy' emotions with 1,566 correct predictions. However, other emotional states had far fewer correct identifications, signalling a predictive bias towards 'happy' expressions.

Conversely, the 'Disgust' category had only 58 correct predictions. This suboptimal performance is attributable to its underrepresentation in the training dataset.

Additionally, some misclassifications were observed. For example, 296 'sad' images were incorrectly classified as 'neutral,' and 183 'fear' images were misclassified as 'sad.'

## Performance analysis

Figure 11 shows the metrics that determine the training and validation loss as described below.

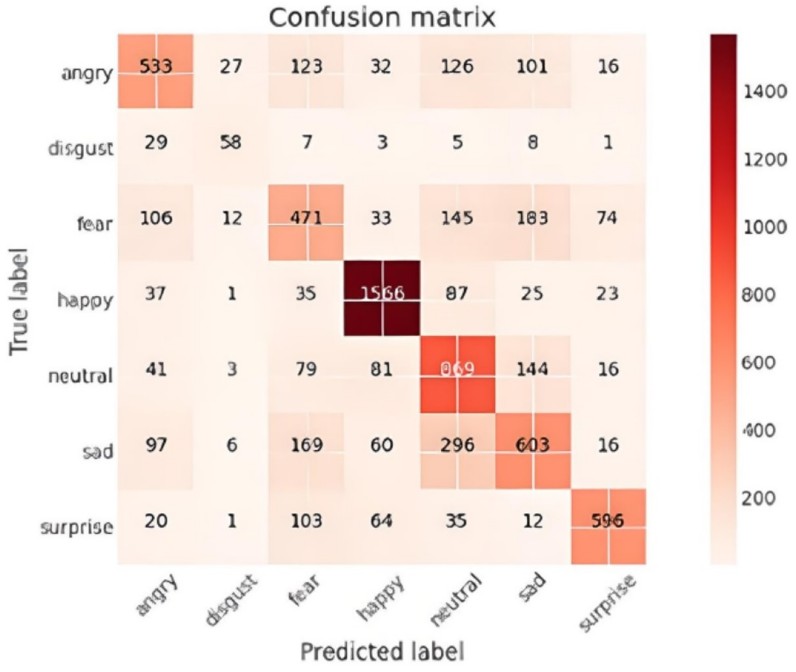

**Figure 10  Confusion matrix.**               

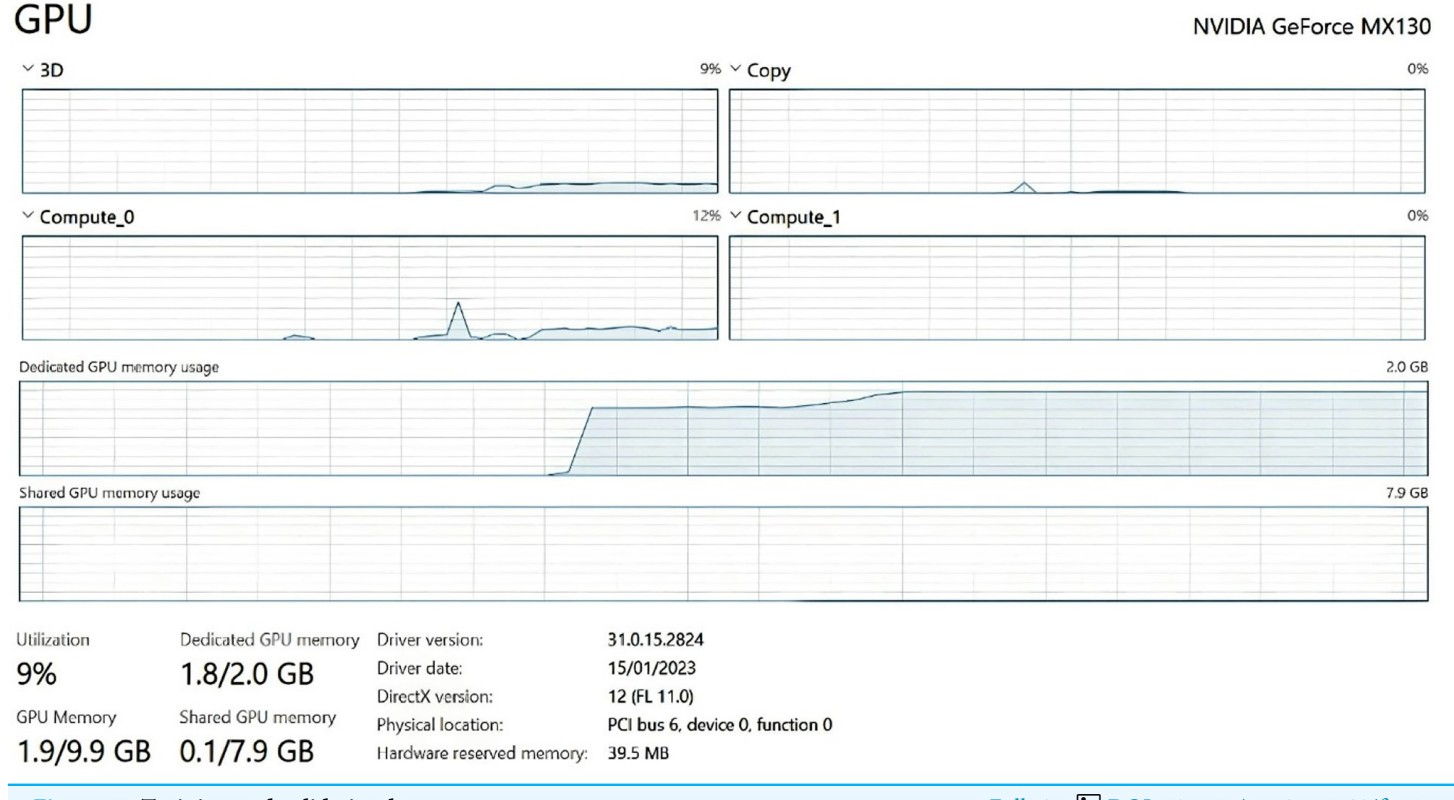

**Figure 11  Training and validation loss.**               

Computational resources: Training the model on a single NVIDIA GeForce MX130 GPU took approximately 12 h for around 100 epochs with an image batch size of 32, an Adam optimizer with a learning rate of 0.001, and a categorical cross-entropy loss function. The computational power required for both model training and detection was deemed reasonable, meaning that the model can be trained and utilized on relatively standard and affordable GPUs that are easily accessible to most users.

Processing speed: Processing speed was evaluated by measuring the time taken for the model to predict the emotional label of an uploaded image. The average inference time was 50 milliseconds for processing an image on a single GPU. This processing speed would enable real-time or near-real-time emotion detection making it suitable for real-world application.

Memory usage: The FER model requires approximately 1.8 GB of GPU memory during real-time emotion detection. The large GPU memory requirement can be attributed to the complexity and size of the model's architecture. The memory usage of the model during training was higher and used up all of the dedicated GPU memory due to the additional overhead of gradient computations and optimization algorithms.

## Discussion of model effectiveness for ASD intervention

The qualitative effectiveness of the implemented model was inferred in two ways, first was through contextual consideration, which principally focused on whether the FER software is in alignment with the unique qualities and needs of individuals with ASD. Second, quantitative measures obtained during performance analysis of the model (such as model accuracy, and loss) were also utilized to provide additional insight that further supports the assessment.

## Contextual considerations

The software implemented used a strength-based approach for emotional training, rather than being focused on the limitations that people with ASD may exhibit.

As such, the model was designed with various features in mind that align with the cognitive capabilities of individuals on the spectrum to yield the maximum benefits from the training process. This included providing visual aids in the form of images and videos to depict facial expressions, therefore catering to the visual learning style typical of ASD patients. Visual cues were also kept consistent to tailor to the community's preference for predictability and structured environments. Systematic feedback in the form of bar charts that display emotional state predictions, as well as on-face labelling of emotions, would reinforce learning by leveraging the unique pattern recognition and systemization skills characteristic in individuals with ASD.

Furthermore, the software was designed particularly with usability in mind. The user interface kept sensory distractions at a minimum, presented an intuitive and straightforward design, as well as used visual and auditory cues. All design features are well-tolerated by individuals with sensory sensitivities.

Although the aforementioned factors may make the software effective for intervention, further considerations need to be taken. For instance, it may be necessary to include

customizability features in future versions to accommodate individual differences in FER needs and preferences, since no "one size fits all" approach applies to ASD. Moreover, whilst valuable insights were obtained from the development-reported qualitative assessment, effectiveness may be subject to bias. This could be remedied by obtaining direct feedback from the target audience of the system, as well as their clinicians and caregivers, and using their perspective for system enhancement.

## Quantitative measures

The processing speed of 50 ms is considered relatively fast for FER, which may aid in providing timely feedback during the training process, or if the system is implemented in wearable technology to be used in day-to-day encounters. However, whilst the FER model can correctly recognize emotions from facial expressions with some level of accuracy, a rate of 66% indicates that in around one-third of cases, the system is subject to emotion misclassification, reducing software effectiveness. Furthermore, the performance analysis presented in 5.3 showed that GPU utilization was high, at 2 GB of memory used for emotional inference. Since computational resources needed to run the program are relatively high, usability may be restricted in resource-constrained environments, which may constrain some individuals who have lower-end computers from benefitting from the solution. Slow responses due to slower processing, as well as inaccurate predictions may also impact engagement levels of users as they use the system for training.

The advantage of this study is not only applying deep learning models for ASD, but additionally this model was integrated with a user interface, carefully tailored for high-functioning autistic adults, which complements the system's real-time streaming and media versatility, which are considered a comparable features that were not included in the previous studies applied in *Talaat (2023)*, *Dhuheir et al. (2021)*, *Li & Wang (2018)*.

Overall, there is a need for improvement in terms of model accuracy and other performance metrics to ensure that emotional recognition is reliable and suitable for use as an intervention method. Enhancement strategies, such as fine-tuning the model architecture, optimizing training parameters, and incorporating additional training data should be deliberated to address model misinterpretations and increase effectiveness.

## CONCLUSION

The SENSES-ASD system represents a significant leap in using machine learning technologies to augment social comprehension among individuals with autism spectrum disorder (ASD). Powered by a convolutional neural network (CNN), the system reached an encouraging yet improvable facial emotion recognition accuracy of 66%. Despite its early promise, the model also highlighted areas for refinement, such as overfitting and class biases. A particular strength is the user interface, carefully tailored for high-functioning autistic adults, which complements the system's real-time streaming and media versatility. These features underscore the system's practicality as both an educational instrument and a stepping stone to improved social interactions for those with ASD. Future work aims to elevate this foundation by adopting multi-modal sensing techniques that merge visual and auditory cues, exploring the use of unsupervised learning for more individualized emotion

recognition, and leveraging time-series analytics to create more accurate predictive emotional models. Overall, the SENSES-ASD system sets the stage for an increasingly nuanced and reliable tool that can benefit broader ASD interventions.

### Funding
This work is funded by the Deanship of Scientific Research at Imam Mohammad Ibn Saud Islamic University (IMSIU) through Research Partnership Program no RP-21-07-09. The funders had no role in study design, data collection and analysis, decision to publish, or preparation of the manuscript.

### Grant Disclosures
The following grant information was disclosed by the authors:
Deanship of Scientific Research at Imam Mohammad Ibn Saud Islamic University (IMSIU) through Research Partnership Program: RP-21-07-09.

### Competing Interests
Tawfik Al-Hadhrami is an Academic Editor for PeerJ.

### Author Contributions
- Haya Abu-Nowar conceived and designed the experiments, performed the experiments, analyzed the data, performed the computation work, prepared figures and/or tables, and approved the final draft.
- Adeeb Sait conceived and designed the experiments, performed the experiments, analyzed the data, performed the computation work, prepared figures and/or tables, and approved the final draft.
- Tawfik Al-Hadhrami conceived and designed the experiments, performed the experiments, performed the computation work, prepared figures and/or tables, authored or reviewed drafts of the article, and approved the final draft.
- Mohammed Al-Sarem conceived and designed the experiments, performed the experiments, authored or reviewed drafts of the article, and approved the final draft.
- Sultan Noman Qasem conceived and designed the experiments, performed the experiments, prepared figures and/or tables, authored or reviewed drafts of the article, and approved the final draft.

### Data Availability
The SENSES-ASD-CNN data and code are available at Zenodo: Sait, A. (2023). SENSES-ASD-CNN [Data set]. Zenodo. https://doi.org/10.5281/zenodo.10223688.
The FER-2013 Dataset is available at Kaggle: https://www.kaggle.com/datasets/deadskull7/fer2013.

## Supplemental Information

Supplemental information for this article can be found online at http://dx.doi.org/10.7717/peerj-cs.1792#supplemental-information.

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
