# Peer review of "SENSES-ASD: a social-emotional nurturing and skill enhancement system for autism spectrum disorder"

_PeerJ Computer Science, doi:10.7717/peerj-cs.1792_

## Round 0.1 · original submission · Minor Revisions

Thank you for your submission, Dear authors,
your manuscript needs improvement in the following aspects

The Social-Emotional Nurturing and Skill Enhancement System is a viable and ground-breaking method for supporting people with Autism Spectrum Disorder (ASD), according to the research (SENSES-ASD). With an emphasis on enhancing social interaction and communication utilizing deep learning technologies, notably convolutional neural networks, this study tackles a crucial aspect of ASD intervention (CNNs).

This study's use of deep learning, particularly convolutional neural networks, to simplify facial emotion recognition is a remarkable strength. This strategy could greatly improve the ability of people with ASD to understand and express their emotions. In addition, the system's adaptability in accepting different media inputs is notable since it enables a more thorough assessment of emotional states.

It might be advantageous to include some particular quantitative outcomes or performance measures to make the evaluation more educational. The work's credibility would be enhanced and readers' comprehension of the relevance of the findings would be improved by providing information on the accuracy levels attained or any comparative analysis with existing approaches.

The paper's effect and applicability in addressing the requirements of people with ASD would be strengthened by additional discussion of the system's performance indicators and potential real-world applications.

Please update the abstract to add more details about the methodology and the results.

A comparison/validity of the study could be more beneficial.

**Language Note:** PeerJ staff have identified that the English language needs to be improved. When you prepare your next revision, please either (i) have a colleague who is proficient in English and familiar with the subject matter review your manuscript, or (ii) contact a professional editing service to review your manuscript. PeerJ can provide language editing services - you can contact us at [email protected] for pricing (be sure to provide your manuscript number and title). – PeerJ Staff

Reviewer 1 ·

Basic reporting

1. In the abstract it seems that author/s have described his/her own technique in full but never discussed the constraints for which this contribution has been made. Please state the novelty for which this work has been proposed.

2. The related work is well written but I will suggest the author to include two tables such that one table describe the previous research contributions from the beginning and may summarize the entire historical background related to the topic. Whereas the table 2 describes the constraints related to the state of the art approaches specially from 2018 to 2023.

3. FER System flow diagram should be more detailed in terms of illustration and explanation as well.

4. Figure 5,10, 11, 14 and 15 are very blur

5. Conclusion must be improved.

Experimental design

No Comment

Validity of the findings

6. Results are not enough I suggest to add more results along with technical analysis. Please be more exhaustive while explaining the results.

7. Can you do the comparison of your results with other results in tabular form? It will be really informative and can attract maximum readers.

Additional comments

8. Divide the literature review into two parts. One is historical work and second one within should be state of the art approaches.

9. Please add some more results. If you can make a GUI for this work it will be more attractive.

10. Please read the paper again to remove some minor grammatical and punctuation errors.

·

Basic reporting

The manuscript demonstrates clear, unambiguous, professional English throughout, making it easy to understand and follow the content. The language used is appropriate for an international audience, ensuring that readers can comprehend the text effectively.

The literature references are well-incorporated, and the manuscript provides sufficient field background and context, which helps to establish the relevance and significance of the research. This demonstrates a strong foundation of knowledge and understanding in the field.

The article structure is professional and adheres to the standards expected in academic publishing. The figures and tables included are relevant, high quality, and well-labeled, enhancing the clarity and presentation of the findings. Additionally, the authors have shared the raw data, which is commendable and allows for transparency and reproducibility.

However, it is important to note that the formal results section should include clear definitions of all terms and theorems, as well as detailed proofs. This will strengthen the validity and rigor of the findings, ensuring that readers can fully comprehend and evaluate the results.

Overall, the manuscript demonstrates strong qualities in terms of language, literature references, structure, figures, tables, and sharing of raw data. By addressing the suggestion regarding the formal results section, the manuscript will further enhance its clarity and validity.

Experimental design

The original primary research conducted in this study demonstrates a commendable level of quality and relevance. The research question is well-defined, relevant, and meaningful, addressing a specific knowledge gap in the field. The study also showcases a rigorous investigation that adheres to high technical and ethical standards, ensuring the reliability and validity of the findings. Furthermore, the methods employed in the research are described with sufficient detail and information, allowing for replication and further exploration of the topic. Overall, this study exemplifies the standards expected in original primary research within the aims and scope of the journal.

Validity of the findings

The manuscript demonstrates a strong adherence to scientific rigor by providing robust and statistically sound underlying data. The inclusion of all data sets ensures transparency and allows for meaningful replication of the study. The conclusions are well-stated and effectively linked to the original research question, providing support for the results obtained. This approach enhances the credibility and reliability of the findings, contributing to the overall quality of the manuscript.

Additional comments

The manuscript demonstrates a commendable effort in addressing the important issue of emotion recognition in individuals with autism spectrum disorders (ASD). The authors have compiled an extensive data set through detailed fieldwork, which adds strength to their research findings. Additionally, the manuscript is well-written in professional and unambiguous language, making it easily understandable for a wide audience. The inclusion of real-life applications, such as incorporating the emotion recognition system into wearable devices, showcases the potential impact and novelty of this research. Overall, the manuscript presents a valuable contribution to the field of emotion recognition and its implications for individuals with ASD.

---

## Round 0.2 · accepted · Accept

Thank you for your fine contribution, the experts are happy to accept your revised paper

Reviewer 1 ·

Basic reporting

All ok

Experimental design

All ok

Validity of the findings

Things have been improved by the authors as per my provided comments.

Additional comments

All authors have already addressed all of my concerns in well manner. Thus, I recommend this paper to be accepted in this present form.

·

Basic reporting

The author has responded to all comments and it is ready for publication. I am happy to accept this paper for publication.

Experimental design

The author has responded to all comments and it is ready for publication. I am happy to accept this paper for publication.

Validity of the findings

The author has responded to all comments and it is ready for publication. I am happy to accept this paper for publication.

Additional comments

The author has responded to all comments and it is ready for publication. I am happy to accept this paper for publication.